# Valley-locked waveguide transport in acoustic heterostructures

Mudi Wang [1], Wenyi Zhou[1], Liya Bi[1], Chunyin Qiu [1], Manzhu Ke [1] & Zhengyou Liu [1,2]✉

Valley pseudospin, labeling the pair of energy extrema in momentum space, has been attracting attention because of its potential as a new degree of freedom in manipulating electrons or classical waves. Recently, topological valley edge transport of sound, by virtue of the gapless valley-locked edge states, has been observed in the domain walls of sonic crystals. Here, by constructing a heterostructure with sonic crystals, a topological waveguide is realized. The waveguide states feature gapless dispersion, momentum-valley locking, immunity against defects, and a high capacity for energy transport. With a designable size, the heterostructures are more flexible for interfacing with the existing acoustic devices than the domain wall structures. Such heterostructures may serve as versatile new devices for acoustic wave manipulation, such as acoustic splitting, reflection-free guiding and converging.

---

[1] Key Laboratory of Artificial Micro- and Nano-structures of Ministry of Education and School of Physics and Technology, Wuhan University, 430072 Wuhan, China. [2] Institute for Advanced Studies, Wuhan University, 430072 Wuhan, China. ✉email: zyliu@whu.edu.cn

Topological edge states, which are well known for gapless dispersion and the resulting one-way transport, have been extensively studied in electronic[1–8], photonic[9–14], and acoustic[15–31] systems in recent years. Such states originate from the so-called bulk-boundary correspondence, rooted in the non-trivial Chern number of materials. Particularly, for two-dimensional valley Hall insulators, the nontrivial valley Chern number gives rise to topological valley edge states[4,5,12,13,25–30]. The valley edge states exist at the interface of two valley phases with opposite valley Chern numbers or band inversion, which are gapless and momentum-valley locked. Recently, valley-locked edge states with one-way transport have been demonstrated in sonic crystals (SCs)[27]. Here, by introducing a heterostructure consisting of SCs with valleys[32–35], a topological waveguide is achieved. Analogous to the valley-locked edge states, the wave-guide states here are also gapless, valley locked, and reflection free but provide high flexibility in manipulating acoustic waves using the width degree of freedom, which opens a unique route toward acoustic guiding, splitting, and converging, advancing the current knowledge on the control of acoustic waves[36–42].

## Results

### The $A\,|\,B_x\,|\,C$ heterostructure.

We consider a sandwich-like structure consisting of three domains, A, B, and C, as shown in Fig. 1a. Compared with the domain wall structure A | C with only A and C domains previously considered[4,5,12,13,25–30], the main difference is the introduction of the B domain of $x$ layers between

A and C. Each domain is an SC consisting of a hexagonal array (33.8 mm lattice constant) of regular triangular rods (20.0 mm sides), with the orientation described by the rotation angle α. For A, B, and C, α is 30°, 0°, and −30°, respectively. In the experimental implementation, the rods have a limited height of 12.2 mm and are arrayed inside a planar air waveguide with axes perpendicular to the top and bottom plates (see Methods). Figure 1b shows that a bandgap exists between the first and second bands for α = 30°, while it is closed for α = 0° with the emergence of a Dirac degeneracy at frequency $\omega_D$ at K (and K'); the bandgap reopens for α = −30°, with the two bands inverted[27,30], as indicated by the '+' and '−' signs. The extrema pair at K and K' of each band are known as the valleys (pseudospin), and A, B, and C are all referred to as valley SCs. Acoustic waves can propagate in the single SC B in the frequency region prohibited in A and C by the bandgap. The heterostructure is illustrated in Fig. 1a, which is a waveguide when the frequency falls in the bandgap of A and C. In Fig. 1c, the left panel gives the projected band structure corresponding to Fig. 1a along $k_X$, obtained by simulations with COMSOL Multiphysics (see Methods). We can see that a gapless band, indicated by the green line, traverses at K and K' the entire bulk gap of A and C, delimited by the two horizontal dashed lines. This band shares great similarity to that of the valley edge states in the domain wall structure A | C: it is gapless and momentum-valley locked with a positive group velocity at K and a negative group velocity at K'. In contrast to the valley edge states[4,5,12,13,25–30] existing at the A | C boundary, here, the valley-locked guiding states extend into the entire B domain, as shown

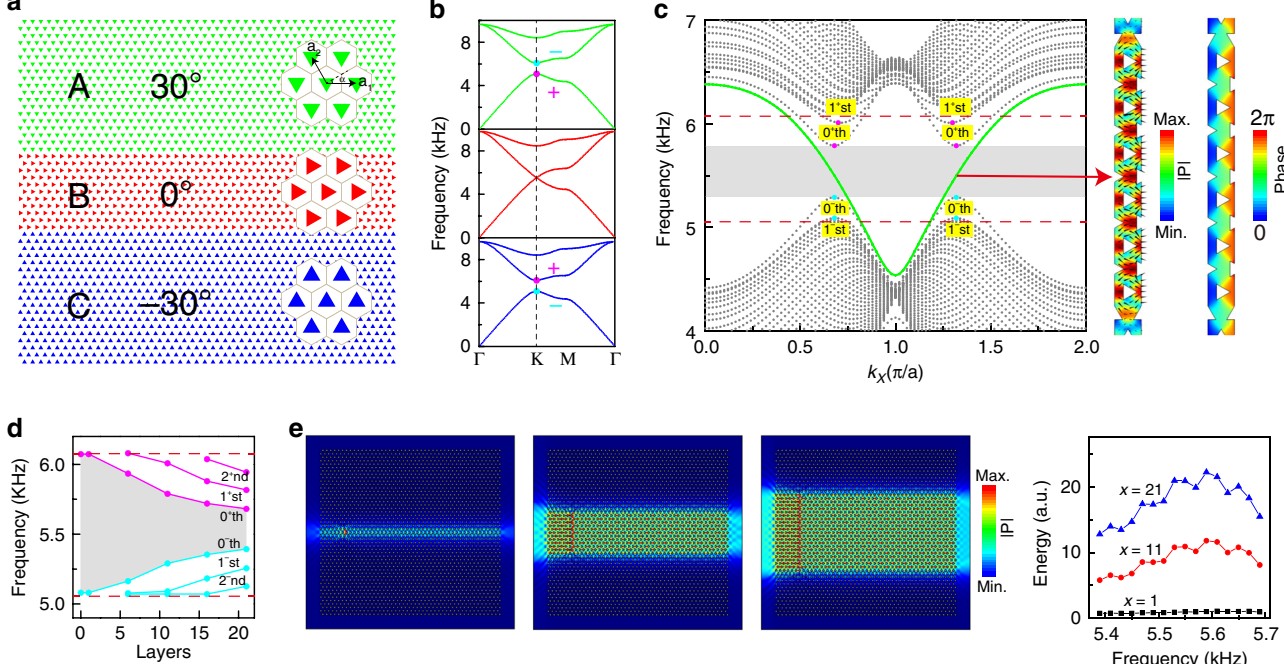

**Fig. 1 Heterostructure $A\,|\,B_x\,|\,C$, with $x$ being the number of layers in domain B, and the topological valley-locked waveguide states. a** Schematic of A | $B_x$ | C structures consisting of three SC domains, A, B, and C. Each SC is an array of triangular rods in a hexagonal lattice with base vectors $\mathbf{a_1}$ and $\mathbf{a_2}$. The rotation angle α defines the orientation of the triangular rods with respect to $\mathbf{a_1}$, which is 30° for A, 0° for B, and −30° for C. **b** Band diagrams for A, B, and C. **c** Left panel: projected band structure for A | $B_{11}$ | C. The shadowed area in the bulk gap (delimited by the two red dashed lines) of domain A or C marks the region in which the TVWSs, indicated by the green line, exclusively exist. The 0th and 1st bands are the other gapped waveguide states. Right panel: absolute pressure | P | (colors) and energy flux (arrows) fields, and the pressure phase (colors) field of the topological waveguide states in domain B. **d** Variation in the width of the shadowed region with $x$, the number of layers in B describing the width of the B domain. The cyan (magenta) lines or dots mark the lower (upper) boundary of the 0th, 1st, and 2nd bands. **e** Left panel: simulated pressure fields at a frequency of 5.55 kHz in the A | $B_x$ | C structures with $x$ = 1, 11, and 21; line sources, marked in red and mimicked by a line of 2, 12, and 22 discretized point sources, respectively, each in the layer separations, are perpendicularly placed inside the waveguides to excite the modes. Right panel: calculated total transmitted energy versus frequency for the three structures.

in the right panel of Fig. 1c by the field distributions of the pressure amplitude, energy flux and pressure phase. We refer to this band as the topological valley-locked waveguide states (TVWSs). In addition to this band, other guiding wave bands, denoted by $0^+$th, $0^-$th, $1^+$st and $1^-$st, can also be seen in Fig. 1c. However, all these bands are gapped and thus lack the one-way feature of the TVWSs. The gap between the $0^+$th and $0^-$th bands, marked by the shadowed area in Fig. 1c, constitutes an exclusive passing frequency window for the TVWSs, which is important for studying and utilizing the novel TVWSs. When the width of the waveguide or the number of layers $x$ in B varies, the passing frequency window changes, as depicted in Fig. 1d. The passing frequency window, marked by the shadowed area, narrows with increasing $x$, and higher-order guiding states enter the bulk gap one by one (see Fig. 1d and Supplementary Fig. 1). The window can be inferred to finally close when the waveguide width sufficiently increases, and the waveguide bands are restored to the bulk band of B. As noted, these guiding states are bisected into two branches, the plus (+) and minus (−) branches, which obviously originate from the unique Dirac cone dispersion of B (see more details in Supplementary Note 1).

To physically determine the origin of the TVWSs in $A\,|\,B_x\,|\,C$, let us examine the properties of crystals A, B, and C. With the **k·p** perturbation method, we know[27] that crystals A, B, and C, identified by the orientation angle $\alpha$, can all be described by a Dirac Hamiltonian, around valley K, referring to the Dirac frequency $\omega_D$ at K, $\delta H = c_D k_X \sigma_X + c_D k_Y \sigma_Y + m c_D^2 \sigma_Z$, in the Hilbert space spanned by $\left(\psi_+^0, \psi_-^0\right)$, the two degenerate Dirac states in the crystal with $\alpha = 0$, or crystal B. $c_D$ is the Dirac velocity or the slope of the linear Dirac cone, $\sigma_i (i = X, Y, Z)$ are the Pauli matrices, and $m$ is the Dirac mass, with $m < 0$ for A, $m = 0$ for B, and $m > 0$ for C. Note that $k_i (i = X, Y, Z)$ are measured from K. The Dirac Hamiltonian at valley K' is simply the time reversal of that at K. The band inversion between A and C actually corresponds to sign reversal of the Dirac mass for the two crystals. From the eigenvalue equation $\delta H \phi = \delta \omega \phi$, $\delta^2 \omega = c_D^2 \left(k_X^2 + k_Y^2\right) + m^2 c_D^4$ can be derived, which describes the dispersion relation of the crystals; for B, this relation gives a Dirac cone since $m = 0$, whereas for A and C, it gives two gapped bands with a separation of $2 m c_D^2$ since $m \neq 0$. Consider the heterostructure $A\,|\,B_x\,|\,C$. Suppose that the thickness of B is $L$, the $A\,|\,B$ interface is located at $Y = L/2$ and the $B\,|\,C$ interface is located at $Y = -L/2$. For convenience, we assume that the Dirac mass of C is $m = M > 0$; then, for A, $m = -M < 0$. The waveguide state, $\phi_{ABC}$, if it exists, should exponentially attenuate along the $+Y$ direction in A and along $-Y$ in C, which requires $k_Y = i\sqrt{c_D^2 k_X^2 + m^2 c_D^4 - \delta^2 \omega}/c_D$ in A with $Y > L/2$ and $k_Y = -i\sqrt{c_D^2 k_X^2 + m^2 c_D^4 - \delta^2 \omega}/c_D$ in C with $Y < -L/2$. Although the problem is difficult to fully solve, a specific solution with $\delta \omega = c_D k_X$ and $\phi_{ABC} = (1, 1)^T$, or in coordinate representation

$$\phi_{ABC}(X, Y) = \left(\psi_+^0 + \psi_-^0\right) \begin{cases} e^{ik_X X - Mc_D(Y - L/2)}, & Y > L/2 \\ e^{ik_X X}, & -L/2 \leq Y \leq L/2 \\ e^{ik_X X + Mc_D(Y + L/2)}, & Y < -L/2 \end{cases} \quad (1)$$

can easily be identified, which satisfies $\delta H \phi = \delta \omega \phi$ in all the A, B, and C regions and is continuous at the boundaries. This analytically demonstrates the existence of the TVWSs. With the same gapless linear dispersion at the K valley as the edge states in $A\,|\,C$, the waveguide states can only propagate along $+X$ due to the positive slope, consistent with the above numerical results. Note that the velocity of the TVWSs, $c_D = 194.8 \ m/s$, or the slope of the dispersion relation $\delta \omega = c_D k_X$, is the same as that of the

bulk waves in B ($\delta \omega = \pm c_D k$). Therefore, crystals A and C with band inversion or Dirac mass reversal and crystal B with the Dirac cone dispersion, through the inherent connection in their crystalline structures, together account for the existence of the TVWSs. As a simple picture, the TVWS in $A\,|\,B_x\,|\,C$ can be understood to be the result of the edge state in $A\,|\,C$, combining a bulk state of B between A and C. These states are compatible because they share the same velocity. This picture also explains why the dispersion of waveguide states, $\delta \omega = c_D k_X$, is independent of the thickness of B. Therefore, the TVWSs can also be understood to appear in $C\,|\,B_x\,|\,A$ (Supplementary Fig. 3), but with a dispersion of $\delta \omega = -c_D k_X$ near the K valley, which can be proved similar to above. This means that in the case of $C\,|\,B_x\,|\,A$, the TVWSs locked to K propagate along the $-X$ direction, in contrast to the case of $A\,|\,B_x\,|\,C$. However, if B is replaced by air, i.e., $A\,|\,air\,|\,C$ (Supplementary Fig. 2d–f), then no such waveguide state can exist because the wave velocity in air $c = 349 \ m/s$, does not match $c_D = 194.8 \ m/s$, although other waveguide states can be available. Additionally, in heterostructure $A\,|\,B_x\,|\,A$ (Supplementary Fig. 2a–c) or $C\,|\,B_x\,|\,C$, TVWSs cannot exist either, according to the above analysis (see more details in Supplementary Note 1).

Obviously, a waveguide with a large width, or $x$, has a high capacity for energy transport, which can be demonstrated by numerical simulations. As illustrated in the left panel of Fig. 1e, line sources, marked in red, are perpendicularly placed inside three waveguides, with $x = 1$, 11, and 21, to excite the TVWSs. Considering the discontinuity inside the crystals, the line sources are mimicked by a line of discretized point sources, 2, 12, and 22, respectively, each located in the layer separations. Here, the uniform distributions of the sources ensure an equal per area energy supply for all three waveguides so that the transmitted energies can be compared. The pressure amplitude distributions inside the three waveguides are calculated and plotted, which clearly show the guiding states propagating inside B. The total transmitted energy, obtained by integrating the energy flux over the right side of the waveguides, increases with increasing $x$, as shown in the right panel of Fig. 1e. Therefore, the TVWSs are more flexible for use in applications than the valley edge states in the $A\,|\,C$ structure.

**Observation of the TVWSs.** We further performed experiments to confirm the TVWSs. Figure 2a shows a photo of the sample with $x = 11$. Acoustic waves are guided into the sample at the point labeled by a star on the left side using a small rubber pipe. As the opening of the pipe is very small compared with the working wavelengths in the experiment, it can be viewed as a point source. Fixing the excitation frequency at 5.55 kHz, the pressure field distribution (containing both the amplitude and phase information) in the rectangle delineated by the black dashed lines is measured (see Methods) and Fourier transformed, and the Fourier spectrum is shown in Fig. 2b by color. We note that only the K valley is highlighted, indicating that the excited TVWS, propagating rightward, is K-valley locked. This is consistent with the dispersion relation of the TVWSs in Fig. 1c, showing a positive group velocity at K. By changing the excitation frequency and measuring and Fourier transforming the pressure field along the horizontal black dashed line, the entire dispersion curves for the rightward waveguide states can be mapped out, as plotted in Fig. 2c by color. Good agreement with the numerical dispersions, depicted by the solid lines with green for the TVWSs and blue for the 0th guiding states, can be observed.

**Properties of the TVWSs.** Valley-locked (or -polarized) edge states are known to result in an intriguing splitting effect[27,30]. Similar splitting behavior for the TVWSs can be demonstrated.

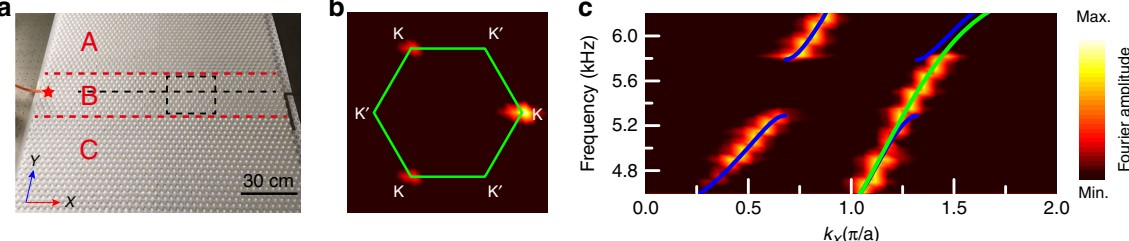

**Fig. 2 Experimental measurement of TVWSs in the waveguide with $x = 11$. a** Photo of the sample, with the red star indicating the point source, operated by opening of a small rubber pipe, which injects the acoustic waves. **b** Fourier spectrum by color at a frequency of 5.55 kHz, obtained by Fourier transforming the measured field distribution inside the rectangle delineated by the black dashed lines in **a**; the green solid lines indicate the hexagonal first Brillouin zone. **c** Experimentally measured dispersions (colors) compared with the simulated ones: the green line indicates the TVWSs, and the blue lines indicate the other waveguide states.

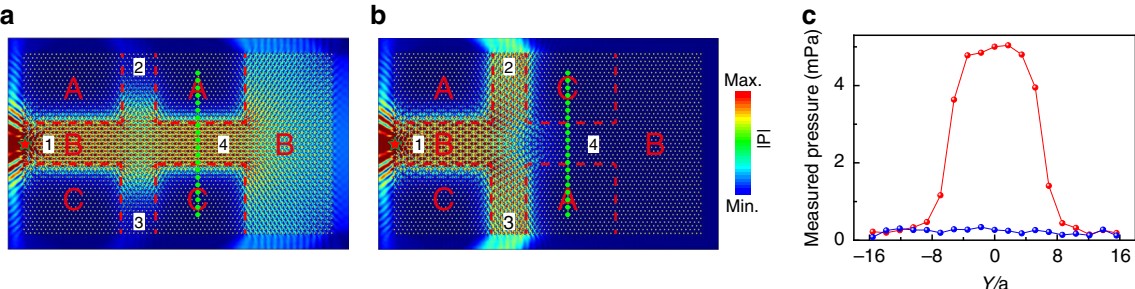

**Fig. 3 Splitting of TVWSs. a, b** Simulated acoustic pressure field distributions under point excitation on the left side for two different configurations at 5.55 kHz. Four input/output ports labeled 1, 2, 3, and 4 are located in domain B. **c** The red and blue dots are the measured pressure profiles along the green dotted lines in **a** and **b**, respectively.

Figure 3a and b shows the configurations of the devices, which consist of a number of A, B, and C domains with borders marked by the red dashed lines. Input and output ports 1–4 are all set in domain B, and port 1, at which a point source is used to excite the TVWSs, serves as the input port. Ports 2 through 4 are treated as output ports. For both devices, the excited TVWSs are locked to the K valley since port 1 is on the left. In the case shown in Fig. 3a, the output from port 2 and port 3 is suppressed because no TVWSs can exist in the channels toward ports 2 and 3 (which are A | B$_x$ | A and C | B$_x$ | C waveguides). The TVWSs can only be transported along the channel toward port 4 (which is an A | B$_x$ | C waveguide, and the TVWSs toward port 4 are also locked to the K valley) and be output. However, interestingly, in the case shown in Fig. 3b, the TVWSs cannot reach port 4 because in this case, the TVWSs in the channel toward port 4 (a C | B$_x$ | A waveguide) are locked to the K' valley and cannot be excited by the TVWSs coming from the input port (which is locked to the K valley). Owing to the symmetry, the TVWSs are bipartitioned into the channels toward port 2 and port 3 (which are both A | B$_x$ | C waveguides, and the TVWSs toward ports 2 and 3 are both K-valley locked). To experimentally show the performance, we measured the pressure profiles along the green dotted lines at port 4 for the two devices in Fig. 3a and b, and the results are depicted in Fig. 3c. The results clearly show that port 4 allows transmission of the acoustic waves in the case of Fig. 3a but forbids it in the case of Fig. 3b, consistent with the simulations.

Due to the valley-locking property, the TVWSs have immunity against defects. To demonstrate this, four different kinds of structural defects, including disorder (Fig. 4a, the rotation angle for the $8 \times 7$ rods within the rectangle delineated by the black dashed lines randomly varies from 0° to 120°), bending (Fig. 4b, domain B is bent by 60°, −120°, and 60° angles in sequence), bulging (Fig. 4c, an area of domain A with $8 \times 7$ rods near domain

B is changed to domain B) and indentation (Fig. 4d, an area of domain B with $8 \times 7$ rods near domain A is replaced by domain A), are deliberately introduced into domain B of the waveguide. Domains A, B, and C in the four cases are delineated by the red dashed lines. A point source, indicated by the red star, is placed on the left side of domain B to excite the TVWSs. At a frequency of 5.55 kHz, the simulated field distributions for the four structures are plotted in Fig. 4a–d by color. We can see that the TVWSs recover after passing through the area with defects. The robustness of the TVWSs against defects can be further demonstrated by experiments. In Fig. 4e, we show the transmitted fields measured at the points marked by green dots near the exit of the four waveguides (Fig. 4a–d), together with that for the waveguide without any defects. We can see that in the passing frequency window indicated by the shadowed area, where only the TVWSs exist, the measured fields are nearly the same for all the cases. However, outside the shadowed region, relatively large differences exist due to the existence of other waveguide or bulk states in domains A, B and C.

Taking advantage of the TVWSs, we can achieve valley-locked and reflection-free converging or focusing of acoustic waves, which may be useful in acoustic enhancement or energy harvesting. Figure 5a and b shows the simulated transport of the TVWSs excited by a point source on the left through a uniform B domain with $x = 21$ and a stepped B domain with $x$ sharply dropping from 21 to 1. For the latter case, almost no reflection occurs when the TVWSs step into the narrow part in B due to the valley-locking properties. This is evidenced by the Fourier spectrum of the field in the area delineated by the black dashed rectangle in the B domain, given in the inset of Fig. 5b, in which only the K valley is highlighted. This means that in this area, only forward-going TVWSs (locked to K) exist, without backward or reflected TVWSs (locked to K'). As a result, the

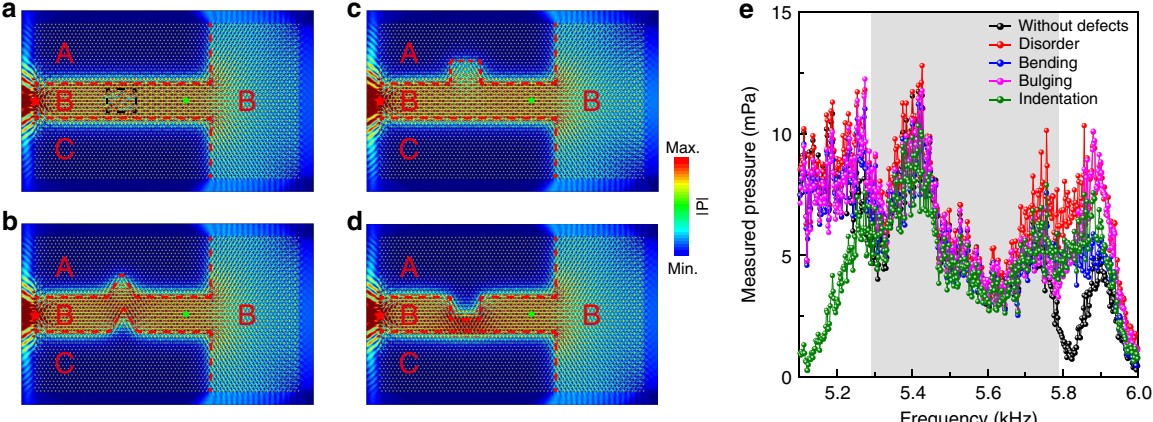

**Fig. 4 Robustness of the TVWSs against defects. a–d** Simulated acoustic pressure field distributions at 5.55 kHz in the waveguides with four different structural defects: disorder, bending, bulging, and indentation. **e** Pressures measured at the points indicated by the green dots near the exits of the four waveguides with defects compared with that of the waveguide without any defects. The good consistency in the shadowed region indicates the immunity of the TVWSs against defects.

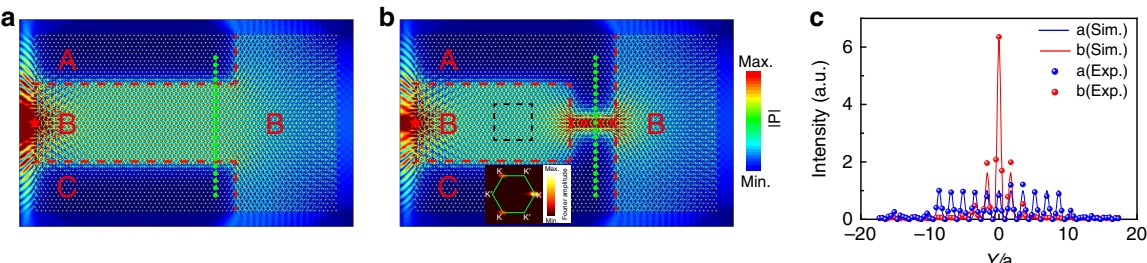

**Fig. 5 Valley-locked converging. a, b** Simulated acoustic pressure field distribution under 5.45 kHz point excitation on the left of the heterostructure with a uniform B domain of $x = 21$ (**a**), and with a stepped B domain with $x$ sharply changing from 21 to 1 (**b**). The A, B, and C domains are delineated by the red dashed lines. The inset in **b** is the simulated Fourier spectrum of the acoustic field in the black dashed rectangle. **c** The blue circles (blue solid line) and red circles (red solid line) represent the experimental (simulated) intensity profiles along the green dotted lines in **a** and **b**, respectively.

acoustic energy converges into the narrow passage in B. To quantitively evaluate the intensity enhancement, we plot in Fig. 5c the intensity profiles along the green dotted lines in Fig. 5a and b, obtained by both simulations and experiments. The reflection-free converging of TVWSs demonstrated here, preserving the valley polarization, may have potential in applications relevant to acoustic information processing.

## Discussion

In conclusion, by designing a heterostructure with three SCs with the valley degree of freedom, a topological waveguide with valley-locked guiding states is obtained. Compared with the two-phase structures hosting the valley edge states, the heterostructures hosting valley-locked waveguide states are more flexible for interfacing with existing devices and applications. Similar to the valley edge states, the valley-locked waveguide states are robust or immune against defects. Unique reflection-free and valley-locked splitting and converging are demonstrated by the hetero-structures. The concept of the topological waveguide in acoustics may inspire analogous research in other systems, such as electronic and photonic systems.

## Methods

**Simulations**. All full-wave simulations are accurately performed by the commercial finite element solver COMSOL Multiphysics. The triangular polymethyl metha-crylate (PMMA) rods used in all structures are modeled as having acoustic rigidity because of their immense impedance mismatch with air. To calculate the projected band structures for various waveguides, such as those in Fig. 1c and Supplementary

Fig. 1, the periodical boundary condition is assumed in the X direction, 49 layers in total for the entire structure (including the x layers for B) are assumed in the Y direction, and rigid boundaries are assumed at the top and bottom. In the simulations of the field distributions, e.g., Fig. 1e and Figs. 3–5, open boundaries are assumed.

**Experimental measurements**. In the experiments, thousands of triangular PMMA rods are arranged in desired configurations within 2D planar air waveguides. The sound signal is transmitted from a narrow tube (diameter ~0.8 cm) and scanned by a movable microphone (diameter ~0.7 cm, B&K Type 4187), and another identical microphone is fixed for phase reference. The phase and amplitude of the pressure field can be obtained by analyzing the acoustic signal with a multiplex analyzer system (B&K Type 3560B).

The sandwich-like sample in Fig. 2 consists of three domains A, B, and C. Domain B includes $11 \times 46$ triangular rods—that is, 11 rows along the Y direction and 46 rods in each row—with rotation angle $\alpha = 0°$. Domains A and C include the same number of rods, that is, $19 \times 46$ rods, but with $\alpha = 30°$ in domain A and $\alpha = -30°$ in domain C. The samples in Fig. 3, Fig. 4, and Fig. 5 all consist of $49 \times 72$ rods.

## Data availability

The data that support the plots within this paper and other findings of this study are available from the corresponding author upon request.

## Code availability

Numerical simulations in this work are all performed using the commercial finite element solver COMSOL Multiphysics. All related codes can be built with the instructions in the Method section.

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

## Acknowledgements

We thank Hua Jiang for discussions. This work was supported by the National Natural Science Foundation of China (Grant Nos. 11804101, 11774275, 11674250, 11534013, and 11747310), the National Key R&D Program of China (Grant No. 2018YFA0305800), and the Natural Science Foundation of Hubei Province (grant number 2017CFA042).

## Author contributions

Z.L. and M.W. conceived the original idea. M.W. performed the simulations. M.W. performed the analytical derivation. M.W., W.Z., L.B., and M.K. carried out the experiments. M.W., W.Z., L.B., C.Q. and Z.L. analyzed the data. Z.L. and M.W. wrote the manuscript. Z.L. supervised the project. All authors contributed to scientific discussions of the manuscript.

## Competing interests

The authors declare no competing interests.
