## [Peer Review File · Nature Communications]

Reviewers' comments:

Reviewer #1 (Remarks to the Author):

The paper presents a combined numerical and experimental study of what the authors call 'topological valley-locked bulk transport in layered phononic crystal'. I do not agree with the choice of words in the title, as well as in the main text, for reasons that I explain below. Overall, the numerics and the experiments are convincing, but their interpretation looks dubious to me. A very deep revision is necessary. Furthermore, the English expression is far from ideal and could be improved a lot at revision.

1. About the choice of words - First, the structure in figure 1a is a 'heterostructure' and it resembles atomic layer deposition samples; I would term the sample $A|B_x|C$ with x the numbers of layers of type B. Compared to the domain wall structures $A|C$ considered before by the authors [28,36], the main difference is the introduction of layer B_x between A and C. The authors should properly acknowledge these facts. I find that the choice of references is particularly inadequate as it piles many citations to papers outside phononics before reaching the really useful ones.

Second, the heterostructure forms a classical waveguide: there are two cladding sections where waves are evanescent in the band gap and a core section where they can be propagating. There is nothing such as 'topological bulk states' in this paper, there are only guided waves!

Third, I would prefer the authors to use 'sonic crystal' rather than 'phononic crystal' because the latter relates to elastic waves, not acoustic waves.

2. The discussion and interpretation misses completely the main result: the fundamental guided wave (green line in fig. 1c) is single-mode whatever the value of x , i.e. the thickness of the (trivial) core. This is at odds with usual waveguides which have an increasing number of modes as the core size increases.

A study of the dispersion relation as a function of the value of x must be made. Fig. 1d starts at $x=1$ but what about $x=0$? How is the dispersion relation changing with x ? Why is the passing frequency closing with increasing x ? Why is there only one guided wave at the band gap center? These are fundamental questions that are not treated in the paper.

3. Technically, there is not enough information on how to reproduce the computations. What super-cell is used for the heterostructure and what are the exact boundary conditions? In their previous papers the authors used periodic super-cells that don't apply here for top and bottom boundaries. What happens if regions A and C are interchanged, i.e. for $C|B_x|A$? The two cases added in the supplemental material are as interesting as that case.

4. The discussion around figures 3, 4, 5 is phenomenological but not sufficiently physical. The word 'topological' is used repeatedly whereas at other parts of the paper it seems that the important thing is the valley-polarization (called vortex Bloch waves in [36]), not a topological property identified by a topological invariant. Actually, the introduction mentions 'insulator', 'semimetal' and 'Chern numbers' but none of these concepts truly apply. I would prefer a more honest discussion of the modal field distributions showing how the vortex waves at the K points distribute; the inset in fig. 1c is too small and blurry to show anything in this respect and all subsequent figures are really too small, without color bars, and seems to erase any phase information.

What is called topological protection may well be the fact that guided mode remains single whatever the core width; that explanation seems to fit all examples given: would the authors agree with that?

Reviewer #2 (Remarks to the Author):

Recently, the topological valley hall effect has been demonstrated in electronics, acoustics, and photonics. (Note that the acoustic one was first realized by several authors of this paper.) However, the full control and use of its intriguing edge states has yet to be achieved. The current work demonstrates the control of multiple domains of valley hall insulators with different valley chern numbers and the realization of beam splitters of valley-projected edge states. The results are interesting, timely, inspiring, and significant to the field of topological

electronics/acoustics/phonics and 2D quantum/meta materials. They may also result in useful applications in electronics/acoustics/phonics devices based on topological valley hall effects.

For these reasons, I strongly suggest the publication of this work in Nature Communications without any delay. Nevertheless, I encourage the authors to consider the following two minor suggestions.

1. Two references are missing. In the same spirit, a very similar valley beam splitter has been realized in electronics recently: "A valley valve and electron beam splitter", Science 362, 1149 (2018). A theoretical work that illustrates valley chern number and topological valley hall effect in various cases (including predicting ref. 5): "Valley Chern numbers and boundary modes in gapped bilayer graphene", PNAS, 110 10546-10551 (2013).

2. I understand that the authors choose the term "topological bulk states" because the B regime is a span of many lattice sites. However, such a regime still serves as a topological boundary (or domain wall), across which the bulk valley chern number switches signs. This is more natural in your cited solid-state experiments in which the domain walls are in fact mesoscopic (\sim a few tens nm).

Reviewers' comments:

Reviewer #1 (Remarks to the Author):

The paper presents a combined numerical and experimental study of what the authors call 'topological valley-locked bulk transport in layered phononic crystal'. I do not agree with the choice of words in the title, as well as in the main text, for reasons that I explain below. Overall, the numerics and the experiments are convincing, but their interpretation looks dubious to me. A very deep revision is necessary. Furthermore, the English expression is far from ideal and could be improved a lot at revision.

Reply: We thank the reviewer for the careful reading of our manuscript and the evaluable comments. According to your constructive suggestions, the title of the manuscript is changed to 'Valley-locked waveguide transport in acoustic heterostructures', and the text is extensively revised and expression is improved to our best.

About the choice of words - First, the structure in figure 1a is a 'heterostructure' and it resembles atomic layer deposition samples; I would term the sample $A|B_x|C$ with x the numbers of layers of type B. Compared to the domain wall structures $A|C$ considered before by the authors [28,36], the main difference is the introduction of layer B_x between A and C. The authors should properly acknowledge these facts. I find that the choice of references is particularly inadequate as it piles many citations to papers outside phononics before reaching the really useful ones.

Reply: Thank you for your valuable comments. According to your suggestion, our sample is now termed as $A|B_x|C$ with x labeling the numbers of layers in B. Particularly, we emphasize in the text that this heterostructure is derived from the domain structure $A|C$ we consider before, by introducing B domain of x layers between A and C domains, to form a waveguide. We also remove the references (original Refs.4,6,7,10,11,14,15,19,20) that are less relevant to phononics.

Second, the heterostructure forms a classical waveguide: there are two cladding sections where waves are evanescent in the band gap and a core section where they can be propagating. There is nothing such as 'topological bulk states' in this paper, there are only guided waves!

Reply: Thank you for the helpful comments and the term 'topological bulk states' throughout the manuscript have now been changed to 'topological waveguide states' or in short 'waveguide states'.

Third, I would prefer the authors to use 'sonic crystal' rather than 'phononic crystal' because the latter relates to elastic waves, not acoustic waves.

Reply: According to your suggestion, 'phononic crystal' has been replaced by 'sonic crystal' throughout the manuscript.

2. The discussion and interpretation misses completely the main result: the fundamental guided wave (green line in fig. 1c) is single-mode whatever the value of x , i.e. the thickness of the (trivial) core. This is at odds with usual waveguides which have an increasing number of modes as the core size increases.

A study of the dispersion relation as a function of the value of x must be made. Fig. 1d starts at $x=1$ but what about $x=0$? How is the dispersion relation changing with x ? Why is the passing frequency closing with increasing x ? Why is there only one guided wave at the band gap center? These are fundamental questions that are not treated in the paper.

Reply: We thank the reviewer for the useful comments and questions which will definitely help improve the manuscript. According to your suggestions, we have given the dispersion relation as a function of x in Fig. R1, including $x=0$ i.e. the case with only A|C domain wall, and we can see that the gaps for $x=0,1$ are nearly equally-sized. Fig. 1d now starts from $x=0$. Just as this reviewer points out, and as shown in Fig. S2f, usual waveguide has usually a number of (rather than single) modes, and the number of modes increases with the increasing of the waveguide width. Consider a usual airborne waveguide of width L and denote the dispersion relation of sound in air as

$= ck$, where $c = 349\text{m/s}$ is the sound speed. At frequency f , $k = \frac{\omega}{c} = \frac{2\pi f}{c}$, and the modes number can be estimated as $n = \text{int}\left(\frac{kL}{\pi}\right) + 1 = \text{int}\left(\frac{2fL}{c}\right) + 1$, where 'int' means taking integer. For $x = 11$, corresponding to a width of the waveguide $L = x \frac{\sqrt{3}}{2} a$, it can be evaluated that $n = 11$ in the bulk gap region of A or C, consistent with that observed in Fig. S2f. Obviously, n increases with L increasing. However, for the waveguide of the heterostructure A|B_x|C, B has the dispersion relation of $\omega - \omega_D = \pm c_D k$ at K and K', where $\omega_D = 2\pi f_D$ with $f_D = 5.515\text{KHz}$ being the frequency at the Dirac point, and k is measured from K or K'. Similarly, at frequency f , $k = \left|\frac{\omega - \omega_D}{c_D}\right| = \frac{2\pi|f - f_D|}{c_D}$ with $c_D = 194.8\text{m/s}$ and the modes number $n = \text{int}\left(\frac{kL}{\pi}\right) + 1 = \text{int}\left[\frac{2|f - f_D|L}{c_D}\right] + 1$. Obviously, for $x=11$ and f falling in the bulk gap of A or C, it can be estimated that $n = 2$, that is, only the zeroth and first orders of modes, for the branch $\omega - \omega_D = ck$, the 0⁺th and 1⁺st, and for the branch $\omega - \omega_D = -ck$, the 0⁻th and 1⁻st, exist, as shown in Fig. S2c and Fig. 1c. For smaller x , for example $x = 6$, it can be evaluated that $n = 1$, which means that in the case only the zeroth modes, i.e. the 0⁺th and 0⁻th can be seen in the gap region. However, with the increasing of the thickness L with $x > 11$, higher orders of modes, e.g. the 2nd order of modes, enter the gap region of A and C, as shown in Fig. R1 here or in Fig. S1 in the supplementary Information.

The zeroth modes of the two branches in waveguide A|B_x|A or A|B_x|C are gapped because of the modulation by the rough walls of A and/or C domains, as can be seen in Fig. S2c and Fig. 1c, indicated by the gray color. However, in the waveguide A|B_x|C, due to the bulk (A and C) and boundary (B) correspondence resulted from the band inversion between A and C, there appears an additional waveguide mode, as shown in Fig. 1c by the green line, which is gapless or topological, traversing the gap center. It is just this mode comprises the start point of this work. The gap of the zeroth modes in A|B_x|C, or the passing frequency for the topological modes, narrows with x increasing as shown in Fig. 1d. This is understandable, because the modulation to the modes, by the walls of A and C, becomes weaker with the waveguide width increasing,

and the gap will finally close to reproduce the Dirac point of B when the waveguide is wide enough. On the contrary, when x is small, the modulation due to the walls is strong, resulting in the wide gap; and when x approaches 0, the gap will finally overlap with the bulk gap, and the zeroth modes merge into the bulk bands of A and C.

Fig. R1 The projected band structures for $x=0,1,6,11,16,21$.

3. Technically, there is not enough information on how to reproduce the computations. What super-cell is used for the heterostructure and what are the exact boundary conditions? In their previous papers the authors used periodic super-cells that don't apply here for top and bottom boundaries. What happens if regions A and C are interchanged, i.e. for C|B x |A? The two cases added in the supplemental material are as interesting as that case.

Reply: Thanks for the questions. Fig. R2a shows the super-cell used the simulations for the heterostructure A|B $_x$ |C with $x=11$. The periodical boundary condition is used in X direction; while in Y direction, there are 49 layers for the whole structure, in addition to x layers for B, and rigid boundaries are assumed on the top and bottom. The commercial software COMSOL can readily handle the simulations for the system. The boundary conditions used for the simulations in the work are now all specified in the Methods. If A and C are interchanged, as shown in Fig. R2b by the supercell, the band structure for the C|B $_{11}$ |A structure is changed to that shown in Fig. R2c. Compared with the band structure for A|B $_{11}$ |C in Fig. 1c, we can see that the topological guiding mode, at K or K', are reversed in propagation direction. We have added this figure into supplemental information.

Fig. R2. **a,b**, The super cells used for the simulations for the heterostructures A|B₁₁|C and C|B₁₁|A respectively in COMSOL Multiphysics. **c**, The projected band structure for C|B₁₁|A.

4. The discussion around figures 3, 4, 5 is phenomenological but not sufficiently physical. The word 'topological' is used repeatedly whereas at other parts of the paper it seems that the important thing is the valley-polarization (called vortex Bloch waves in [36]), not a topological property identified by a topological invariant. Actually, the introduction mentions 'insulator', 'semimetal' and 'Chern numbers' but none of these concepts truly apply. I would prefer a more honest discussion of the modal field distributions showing how the vortex waves at the K points distribute; the inset in fig. 1c is too small and blurry to show anything in this respect and all subsequent figures are really too small, without color bars, and seems to erase any phase information.

What is called topological protection may well be the fact that guided mode remains single whatever the core width; that explanation seems to fit all examples given: would the authors agree with that?

Reply: The phenomena exhibited in Figures 3-5 have all a common topological physical origin, that is, the momentum-valley locking, for the waveguide states in

A|B_x|C structure. In simple A|C structure, momentum-valley locking also occurs for the edge states on the domain wall, referred to as valley edge states specifically (see ref. 4,5,12,13,26-31). This is similar to momentum-spin locking for the edge states in topological insulators, and in fact the valley is also viewed as pseudo-spin. Because of the contrast between valleys K and K' (like spin-up and spin-down), the edge states, locked to a particular valley or spin, can hardly be scattered into those with opposite valley (or spin), resulting in the fascinating topological behaviors, such as unique energy splitting (Fig. 3) and converging (Fig. 5), and defect immune propagating (Fig. 4). The waveguide states or edge states with the topological momentum-valley locking properties are originated from the bulk-boundary correspondence in A|B_x|C and A|C structure due to the band inversion between A and C, the insulators with valleys K and K', and can be identified by the valley Chern numbers of A and C for the valleys (ref. 4,28), serving as the topological invariant. Indeed, the bulk states at the valleys K or K', i.e. the extrema of the bands of A or C, are featured by the vortices in the flux field distributions, but the edge states on the domain wall in-between A and C and the waveguide states in B region here, lose the vortex feature as a result of the coupling of the opposite vortices in A and C (ref. 28). According to your suggestion, in Fig. 1c (as shown below) we give the field distributions, including the amplitude, flux and phase, in the added panels. It is right that the topological protection, is the fact that the guided mode remains single but more importantly, GAPLESS whatever the core width, just as the edge mode of a particular spin in a topological insulator does.

Reviewer #2 (Remarks to the Author):

Recently, the topological valley hall effect has been demonstrated in electronics, acoustics, and photonics. (Note that the acoustic one was first realized by several authors of this paper.) However, the full control and use of its intriguing edge states has yet to be achieved. The current work demonstrates the control of multiple domains of valley hall insulators with different valley chern numbers and the realization of beam splitters of valley-projected edge states. The results are interesting, timely, inspiring, and significant to the field of topological electronics/acoustics/photonics and 2D quantum/meta materials. They may also result in useful applications in electronics/acoustics/photonics devices based on topological valley hall effects.

For these reasons, I strongly suggest the publication of this work in Nature Communications without any delay. Nevertheless, I encourage the authors to consider the following two minor suggestions.

Reply: We thank this reviewer for strongly recommending the publication of our work in Nature Communications. We have carefully considered all the questions/suggestions made, and responded to them below, one by one.

1. Two references are missing. In the same spirit, a very similar valley beam splitter has been realized in electronics recently: "A valley valve and electron beam splitter", Science 362, 1149 (2018). A theoretical work that illustrates valley chern number and topological valley hall effect in various cases (including predicting ref. 5): "Valley Chern numbers and boundary modes in gapped bilayer graphene", PNAS, 110 10546-10551 (2013).

Reply: Thank you. We have added these two references ([8] and [4]) in the revised version of the manuscript.

2. I understand that the authors choose the term "topological bulk states" because the

B regime is a span of many lattice sites. However, such a regime still serves as a topological boundary (or domain wall), across which the bulk valley chern number switches signs. This is more natural in your cited solid-state experiments in which the domain walls are in fact mesoscopic (~ a few tens nm).

Reply: Thank you very much for the informative comments. To be more accurate and also consistent with the cases for classical waves, the states along the 'thick' domain wall is now termed as 'guiding states'.

Reviewers' comments:

Reviewer #1 (Remarks to the Author):

The authors have revised their paper considering my comments. I still find, however, that the presentation and the discussion of the results is not on par with what is expected from a quality journal. I will refer in the following to my initial points.

1. I thank the authors for following the terminology I suggested (heterostructure).
2. The added discussion and figures went to the supplemental material and not to the main text. That could be a good choice were the supplemental text well written and free of ambiguities, but this is not the case. I have the following basic and important objection. Fig. S1 presents the influence of the core size, as I requested, with the crystal phase B as the core material. This crystal B is trivial and this topological feature is used repeatedly in the text to justify the importance and significance of the results. But Fig. S2, with homogeneous air as the core material, is also trivial (!) and obviously the guided modes are not topological in this case. This means that the fact that the core is crystal B and not any other trivial material matters and the discussion of the topological properties of the A|Bx|C and variations falls short. No physical reason is given in the paper why the A|air|C heterostructure does not show the same properties.
3. Thanks for the added information.
4. I just have to repeat my original comment. It is not sufficient to claim many times that this is topological and this is trivial to draw conclusions. The paper is descriptive but the physical explanations are still unsatisfactory.

In addition, I believe the paper needs serious English editing and was not prepared with enough care. The abstract is overselling and not informative. Waveguide topological transport in itself is not new, the A|C domain wall already did it. Actually, the claim that the A|Bx|C heterostructure waveguide transport much more energy than the simple domain wall waveguide is not supported by Fig. 1e: the energy transported only increases by a factor 2 for $x=21$ compared to $x=1$. Overall, the figures can hardly be read when the paper is printed because fonts are much too small.

Reviewer #2 (Remarks to the Author):

The authors have fully addressed all the questions and comments from us. I give my full recommendation now.

Reviewer #1 (Remarks to the Author):

The authors have revised their paper considering my comments. I still find, however, that the presentation and the discussion of the results is not on par with what is expected from a quality journal. I will refer in the following to my initial points.

1. I thank the authors for following the terminology I suggested (heterostructure).
2. The added discussion and figures went to the supplemental material and not to the main text. That could be a good choice were the supplemental text well written and free of ambiguities, but this is not the case. I have the following basic and important objection. Fig. S1 presents the influence of the core size, as I requested, with the crystal phase B as the core material. This crystal B is trivial and this topological feature is used repeatedly in the text to justify the importance and significance of the results. But Fig. S2, with homogeneous air as the core material, is also trivial (!) and obviously the guided modes are not topological in this case. This means that the fact that the core is crystal B and not any other trivial material matters and the discussion of the topological properties of the A|B_x|C and variations falls short. No physical reason is given in the paper why the A|air|C heterostructure does not show the same properties.

Reply: We thank this reviewer for the careful reading of the supplemental text regarding the TVWSs in the heterostructure A|B_x|C. Indeed, crystal B plays an essential role in supporting the TVWSs, while if it is replaced by air, there is no such states. To address this issue, let's have a brief overview to crystals A, B and C first. Commonly, A, B and C are the same hexagonal array of the same triangular rods. The difference for them lies in the orientation of the triangular rods described by α . For B, $\alpha = 0$. A is obtained if the triangular rods in B are rotated 30° anticlockwise or $\alpha = 30^\circ$, while C is derived if they are rotated 30° clockwise or $\alpha = -30^\circ$. Crystal B hosts the Dirac cone dispersion relation $\delta\omega = \omega - \omega_D = \pm c_D k$, as shown in Fig. R1, and it is a 'semimetal'. The Dirac degeneracy in B is lifted when the triangular rods are rotated, therefore a band gap opens between the first and second bands in A

and C, as shown in Fig. R1, and A and C are ‘insulator’. As A and C host the valleys (the frequency extrema) at K (and K’), they are referred to as valley insulators. Although the band structures of A and C look the same, however, their first and second bands are interchanged, in other words, there is a band inversion for the two crystals. More explicitly, the state in the first band of A is the same as the state in the second band of C, labeled by ‘+’; and the state in the second band of A is the same as the state in the first band of C, labeled by ‘-’. A full phase diagram, showing the band edges of the first and second bands of the crystal varying with the orientation angle α , is given in Fig. R2, in which the positions of the crystals A, B and C are indicated. By using the k·p perturbation method, the crystal in the phase diagram, identified by α , can be characterized by an effective Dirac Hamiltonian around K [28], $\delta H = c_D k_x \sigma_x + c_D k_y \sigma_y + m c_D^2 \sigma_z$ in the Hilbert space spanned by the bases (ψ_+, ψ_-) , which are two degenerate Dirac states for the crystal with $\alpha = 0$ or crystal B, where $m = \frac{1}{2v c_D^2} \left(\frac{\omega_+^2 - \omega_-^2}{2\omega_D} \right)$, describing the mass of a Dirac particle, c_D is the Dirac velocity or the slope of the linear Dirac cone, $\sigma_i (i = X, Y, Z)$ are the Pauli matrices, and $\omega_i (i = +, -)$ are the band edges dependent of α as shown in Fig. R2. Note that $k_i (i = X, Y, Z)$ are measured from K. Clearly, for crystal A, $m < 0$ as $\omega_+ < \omega_-$ at $\alpha = 30^\circ$, while for C, $m > 0$ as $\omega_+ > \omega_-$ at $\alpha = -30^\circ$, and particularly for B, $m = 0$ as $\omega_+ = \omega_-$, corresponding to a massless particle. Therefore, the band inversion of A and C corresponds to the Dirac mass sign reversing. For convenience, we assume for A $m = -M < 0$, and for C $m = M > 0$ with M being a positive number. The eigenvalue equation

$$\delta H \phi = \delta \omega \phi, \quad (\text{R1})$$

with $\delta \omega$ being measured from the Dirac frequency ω_D , gives the dispersion curve

$$\delta^2 \omega = c_D^2 (k_X^2 + k_Y^2) + m^2 c_D^4. \quad (\text{R2})$$

Considering the A|C structure, and supposing the domain wall aligned with X direction, and A in the region with $Y > 0$ and C in the region with $Y < 0$. The edge states ϕ_{AC} on the domain wall, if exists, should fulfill $\delta H \phi = \delta \omega \phi$ in both the A and C sides, where $\delta H = c_D k_X \sigma_X + c_D k_Y \sigma_Y + m c_D^2 \sigma_Z$, with $m = -M < 0$ in A and

$m = M > 0$ in C. Assuming the solution $\phi_{AC} = (E_1, E_2)^T$ with E_1 and E_2 being constants and T denoting transposing. As the edge states, ϕ_{AC} must attenuate exponentially in both the $\pm Y$ directions, which requires that k_Y must be pure imaginary with $k_Y^2 = (\delta^2\omega - c_D^2k_X^2 - m^2c_D^4)/c_D^2 < 0$, and explicitly, $k_Y = i\sqrt{c_D^2k_X^2 + m^2c_D^4 - \delta^2\omega}/c_D$ in A with $Y > 0$ while $k_Y = -i\sqrt{c_D^2k_X^2 + m^2c_D^4 - \delta^2\omega}/c_D$ in C with $Y < 0$. With these requirements, Eq. (R1) immediately gives

$$\delta\omega = c_D k_X \quad (\text{R3})$$

and $\phi_{AC} = (E_1, E_2)^T = (1, 1)^T$. In addition, $k_Y = iMc_D$ in A, and $k_Y = -iMc_D$ in C. in the coordinate representation, ϕ_{AC} can be written, $\phi_{AC}(X, Y) = (\psi_+^0 + \psi_-^0)e^{ik_X X + mc_D Y}$, or

$$\phi_{AC}(X, Y) = (\psi_+^0 + \psi_-^0) \begin{cases} e^{ik_X X - Mc_D(Y-L/2)}, Y > 0 \\ e^{ik_X X + Mc_D(Y+L/2)}, Y < 0 \end{cases}. \quad (\text{R4})$$

This confirms the existence of the edge states localized on the domain wall, which has a linear dispersion (thus gapless) with positive slope at the K valley, and thus can only propagate along $+X$ [28]. In this sense, the edge states are referred to as topological (K) valley-locked edge states. From the above deduction, one notes that the edge states are rooted in the Dirac mass reversing or band inversion between A and C, and without it no such state can exist. That is the so-called bulk boundary correspondence. Generally it states that on the boundary of two phases with band inversion or opposite Dirac mass, gapless edge states must appear, so that the twist of the band inversion or Dirac mass reversing between the two phases can be smoothed by bridge of the edge states.

Fig. R1. Schematic of the crystals in the domain C, B and A. Lower panel: the corresponding band diagrams.

Fig. R2. The phase diagram of valley SCs, showing the band edges of the first and second bands of the crystal varying with the orientation angle α .

Now, let's turn to the heterostructure A|B_x|C. Suppose that the thickness of B is L , the A|B interface is located at $Y = L/2$ and the B|C interface is located at $Y = -L/2$. Similarly, a waveguide state, if exists, should fulfill $\delta H\phi = \delta\omega\phi$ in all the A, B and C regions, where $\delta H = c_D k_X \sigma_X + c_D k_Y \sigma_Y + m c_D^2 \sigma_Z$, with $m = -M$ in A, $m = 0$ in B, and $m = M$ in C. For convenience, we assume that the Dirac mass of C is $m = M > 0$; then, for A, $m = -M < 0$. The waveguide state, ϕ_{ABC} , if exists, must attenuate exponentially along $+Y$ direction in A and along $-Y$ in C, which also requires $k_Y = i\sqrt{c_D^2 k_X^2 + m^2 c_D^4 - \delta^2 \omega / c_D}$ in A with $Y > L/2$ while $k_Y = -i\sqrt{c_D^2 k_X^2 + m^2 c_D^4 - \delta^2 \omega / c_D}$ in C with $Y < -L/2$. Though it is difficult to solve for the waveguide states analytically in this case, the number of them can be roughly estimated according to a conventional waveguide, as have shown in our previous reply. Nevertheless, referring to the edge states solved in A|C, a solution of

$$\delta\omega = c_D k_X, \quad (\text{R5})$$

and $\phi_{ABC} = (1,1)^T$ or

$$\phi_{ABC}(X, Y) = (\psi_+^0 + \psi_-^0) \begin{cases} e^{ik_X X - M c_D (Y - L/2)}, & Y > L/2 \\ e^{ik_X X}, & -L/2 \leq Y \leq L/2, \\ e^{ik_X X + M c_D (Y + L/2)}, & Y < -L/2 \end{cases} \quad (\text{R6})$$

can be identified straightforward, which satisfies $\delta H\phi = \delta\omega\phi$ in all the A, B and C regions and is continuous on the boundaries. With the same linear dispersion at K valley as the edge states in A|C, the waveguide states are also gapless, and can only propagate along $+X$ due to the positive slope, and in this sense, we refer to them as the topological valley-locked waveguide states (TVWSs). Our work is just intended to demonstrate the TVWSs by the simulations and experiments. Note that, the velocity of the TVWSs, $c_D = 194.8 \text{ m/s}$, or the slope of the dispersion relation $\delta\omega = c_D k_X$, is the same as that of the bulk waves in B ($\delta\omega = \pm c_D k$). Therefore, crystal B, with its Dirac cone dispersion, indeed matters for the existence of the TVWSs. As a simple picture, the TVWS in A|B_x|C, can be understood as the edge state in A|C, combining a bulk state of B between A and C. They are compatible because they have the same velocity. This picture also explains why the dispersion of waveguide states,

$\delta\omega = c_D k_X$, is independent on the thickness of B. Therefore, it can be understood that if A and C interchanges, the TVWSs also appear in C|B_x|A, but with a reversed dispersion, $\delta\omega = -c_D k_X$ near K valley, which can be proved similarly as above. It means that in the case of the C|B_x|A, the TVWSs locked to K propagate along $-X$ direction, in contrast to the case of A|B_x|C. However, if B is replaced by air, no such waveguide state can exist, because the wave velocity in air $c = 349\text{m/s}$ does not match $c_D = 194.8\text{m/s}$, although numerous other waveguide states are available as shown in Fig. S2f and interpreted in our earlier reply. Even if such waveguide states exist in the case of A|air|C, as many waveguide states coexist and there is no exclusive passing frequency window for the states, there is no way to demonstrate and utilize the topological feature of the states.

3. Thanks for the added information.

4. I just have to repeat my original comment. It is not sufficient to claim many times that this is topological and this is trivial to draw conclusions. The paper is descriptive but the physical explanations are still unsatisfactory.

In addition, I believe the paper needs serious English editing and was not prepared with enough care. The abstract is overselling and not informative. Waveguide topological transport in itself is not new, the A|C domain wall already did it. Actually, the claim that the A|B_x|C heterostructure waveguide transport much more energy than the simple domain wall waveguide is not supported by Fig. 1e: the energy transported only increases by a factor 2 for $x=21$ compared to $x=1$. Overall, the figures can hardly be read when the paper is printed because fonts are much too small.

Reply: Thank you for careful reading of our revised manuscript. According to your suggestion, we have added more **physical explanations**, including the analytical verification of the waveguide states, in the revised version. We removed **redundant emphasis in claiming to be** ‘topological’ or ‘trivial’ in drawing conclusions. The **abstract** has been revised with effort. The **English** throughout the manuscript has been polished by American Journal Experts editing company.

Thank you for your valuable comments on the issue regarding the energy transport by the waveguides. Originally, we use a wide Gaussian beam incident from air to excite the waveguide state. Consider the big impedance mismatch between air and crystal B, the excitation is comparatively not as effective when B is thick. That is why **the energy transported increases** with the width of B not very notably. To overcome the impedance mismatch, and quantitatively evaluate the energy transported, we put a line source into domain B of the waveguide perpendicularly, to excite the TVWSs. Considering the un-continuity in the structure of the crystal, the line sources are mimicked by a line of discretized point sources located in the layer separations between A and C, as shown in Fig. R3. We can see that the energy transported by thick waveguide markedly increases. Fig.1e is thus updated accordingly. In the modified version, the font in all the figures has been increased.

Fig. R3. Left panel: simulated pressure fields at a frequency of 5.55 kHz in the A|B_x|C structures with $x=1, 11$ and 21 ; line sources, marked in red and mimicked by a line of $2, 12$ and 22 discretized point sources, respectively, each in the layer separations, are perpendicularly placed inside the waveguides to excite the modes. Right panel: calculated total transmitted energy versus frequency for the three structures.

In summary, the heterostructure A|B_x|C advances the simple domain wall A|C in at least the following respects: (1) The introduction of domain B endows the waveguide and waveguide states with an additional degree of freedom. (2) As the size can be adjusted, the energy transported by the waveguide can be scaled. (3) With the adjustable width, the waveguide is more flexible for interfacing with existing devices and applications. (4) The waveguide states can be immune against imperfections in clusters such as those in Fig. 4. (5) Taking advantage of the additional width degree of freedom, novel acoustic convergence can be realized, as shown in Fig. 5.

REVIEWERS' COMMENTS

Reviewer #1 (Remarks to the Author):

The authors have very significantly improved their manuscript. They have convincingly answered my different questions. I am especially impressed by the new analytical model giving very simply and beautifully the dispersion of guided modes. It was worth asking the authors to provide the readers with true physical explanations beyond the usual topological gossip. The English and the style have improved significantly as well. I am happy to support the publication of this interesting and insightful paper.

I have only the following minor comments that the authors can or not consider for further improvement.

1. The review paper by Ma et al published last year in Nature Reviews [<https://doi.org/10.1038/s42254-019-0030-x>] uses the valley index in the perturbation Hamiltonian of a Dirac cone (their equation (2); your lines 101-102). I don't think it makes any difference in the dispersion relations you derive, but there is a discrepancy with your formulation that puzzles me.
2. You changed the source exciting the waveguides to avoid the impedance mismatch at the entrance in crystal B and that solves the issue I raised, but coupling into the waveguide will remain an issue in applications since actual sources can hardly be embedded in the crystal.

Reviewer #1 (Remarks to the Author):

The authors have very significantly improved their manuscript. They have convincingly answered my different questions. I am especially impressed by the new analytical model giving very simply and beautifully the dispersion of guided modes. It was worth asking the authors to provide the readers with true physical explanations beyond the usual topological gossip. The English and the style have improved significantly as well. I am happy to support the publication of this interesting and insightful paper. I have only the following minor comments that the authors can or not consider for further improvement.

1. The review paper by Ma et al published last year in Nature Reviews [<https://doi.org/10.1038/s42254-019-0030-x>] uses the valley index in the perturbation Hamiltonian of a Dirac cone (their equation (2); your lines 101-102). I don't think it makes any difference in the dispersion relations you derive, but there is a discrepancy with your formulation that puzzles me.

Reply: Thanks for the question. Eq. (2) in the paper by Ma et al, is actually two equations, one describing the Dirac cone at K , labeled with valley index $+1$, and the other the Dirac cone at K' labeled with index -1 , and the wave velocity is set unity. The two cones are related by the time reversal symmetry, as shown in Fig. R1; therefore, one only needs to consider a single cone. In our manuscript, as stated in line 101, we only concentrate on single valley K , corresponding to taking valley index $+1$ in Ma et al's Eq. (2). Besides, our Hamiltonian includes the case of Dirac cones (when $m=0$), or Ma et al's Eq. (2), and also the case with the Dirac degeneracy lifted or with gap ($m \neq 0$). That is to say, we only solve for the waveguide states around K valley. The waveguide states around valley K' can be obtained straightforward by simply taking the time reversal of those at valley K . This is also the reason that in Supplementary Fig. 1 of Supplementary Information we only plot the waveguide states at valley K , because the waveguide states at valley K' are just the time-reversal of those at K . The full dispersion relation including the waveguide

states at both valleys K and K' is as Fig. 1c. We have added a sentence in the text to stress this point.

Fig. R1

2. You changed the source exciting the waveguides to avoid the impedance mismatch at the entrance in crystal B and that solves the issue I raised, but coupling into the waveguide will remain an issue in applications since actual sources can hardly be embedded in the crystal.

Reply: Thanks for the comments. To put an array of point sources in crystal to excite waves is not a difficult issue actually. The point sources used in our simulation locate in the separations of the triangular rods. The heterostructure, as shown in the photo, Fig. 2a, is a planar structure covered with a large plate. Practically, we can drill small holes on the plate just above the excitation positions, put the point sources (e.g. headphones) into place through the holes and then seal the holes with soft material, e.g. rubber, as illustrated schematically in Fig. R2, where part of the cover plate is removed to show the details inside. In our previous work, we have put four point-sources into the crystal in such a way to excite the waves [Fig. 4, Nature Physics, 14, 30-34(2018)].

Fig. R2